**Data Availability Statement:** All relevant data are within the manuscript.

**Funding:** Giovanni Garippa has received research grants by University of Sassari, Italy. Fondo di Ateneo per la ricerca, annualità 2020. Rep. 2465/

# *In vitro* larvicidal activity of ivermectin and povidone-iodine against *Oestrus ovis*

**Giuseppe D'Amico Ricci** [1,2⊗], **Giovanni Garippa** [3⊗], **Stefano Cortese** [4‡], **Rita Serra** [1], **Francesco Boscia** [5], **Stefano Dore** [6,7], **Antonio Pinna** [6,7‡]*

**1** Department of Biomedical Sciences, University of Sassari, Sassari, Italy, **2** Ospedale Sperino-Oftalmico, SC Oculistica II, ASL Città Di Torino, Turin, Italy, **3** Department of Veterinary Medicine, University of Sassari, Sassari, Italy, **4** SC di Farmacia, Azienda Ospedaliero-Universitaria di Sassari, Sassari, Italy, **5** Department of Basic Medical Science, Section of Ophthalmology, Neuroscience and Sense Organs, University of Bari, Bari, Italy, **6** Department of Medical, Surgical, and Experimental Sciences, Ophthalmology Unit, University of Sassari, Sassari, Italy, **7** UOC di Oculistica, Azienda Ospedaliero-Universitaria di Sassari, Sassari, Italy

⊗ These authors contributed equally to this work.
‡ SC and AP also contributed equally to this work.
* apinna@uniss.it

## Abstract

### Purpose

To assess the *in vitro* larvicidal activity of ivermectin and povidone-iodine (PVP-I) against *Oestrus ovis*, the most frequent cause of external ophthalmomyiasis.

### Methods

L1 *O. ovis* larvae were collected from the nasal boots of sheep slaughtered in local abattoirs and transferred onto Petri dishes containing mucosal tissue (25 larvae/dish). The larvicidal activity of the following formulations was tested: 1% ivermectin suspension in balanced sterile saline solution (BSSS), 1% ivermectin solution in propylene glycol, propylene glycol, 0.6% PVP-I in hyaluronic acid vehicle (IODIM®), and combination of ivermectin 1% solution and 0.6% PVP-I. One mL of each formulation was added to different Petri dishes containing the larvae. The time needed to kill the larvae was recorded.

### Results

893 larvae were tested. The median time needed to kill the larvae was 46, 44, 11, 6, and 10 minutes for Iodim®, ivermectin 1% suspension, propylene glycol, ivermectin 1% solution, and a combination of ivermectin 1% solution with 0.6% PVP-I, respectively. Kaplan-Meyer analysis disclosed that the survival curves were significantly lower in samples treated with ivermectin 1% solution, ivermectin 1% solution + 0.6% PVP-I, and propylene glycol than in samples receiving other treatments or BSSS.

### Conclusion

In this *in vitro* study, ivermectin 1% solution in propylene glycol, ivermectin 1% solution + 0.6% PVP-I, and propylene glycol alone showed a good, relatively rapid larvicidal activity against *O. ovis* larvae. Further experimental and clinical studies are necessary to establish

2020, Prot. 0097985 del 01/09/2020. URL of the University of Sassari: https://www.uniss.it The funder had no role in study design, data collection and analysis, decision to publish, or preparation of the manuscript.

**Competing interests:** The authors have declared that no competing interests exist.

whether, or not, these formulations may be considered as potential candidates for the topical treatment for external ophthalmomyiasis caused by *O. ovis*.

## Introduction

Ophthalmomyiasis is the infestation of the eye with maggots (larvae) of certain flies. *Oestrus ovis*, Diptera:Oestridae, is the most frequent cause of ocular myiasis, particularly in countries with tropical or mild climates [1, 2]. The presence of this pathogen is widely known in the Mediterranean area, where sheep farming is common [1–3].

Based on its location, ophthalmomyiasis is classified into external, internal, and orbital. In the external form, larvae are found on the conjunctiva and eyelid margins, leading to inflammation of the ocular surface. Typical symptoms include tearing, photophobia, foreign body sensation, and pain. Larvae may survive on the ocular surface for several days with consequent worsening of signs and symptoms, usually due to an IgE-mediated immune response to larval antigens.

Currently, mechanical removal of the larvae is the only available treatment for external ophthalmomyiasis. However, this procedure is time-consuming and often not completely effective, because the highly motile larvae can easily be missed. This usually results in persistence of ocular surface inflammation, with the need for repeated consultations.

There are sparse reports describing cases of ophthalmomyiasis caused by other Diptera species (*Dermatobia hominis* and *Cochliomyia hominivorax*) treated with ivermectin (MK-933, 22,23-Dihydroavermectin B1), a semisynthetic anthelmintic agent [4–7]. Oral ivermectin has also been successfully used in the treatment of one case of human conjunctival myiasis caused by *O. ovis* [8]. Furthermore, 0.6% povidone-iodine (PVP-I), an iodinated polyvinyl polymer, has recently been found to be effective against *Demodex mites*, a parasite colonizing the eyelid margins [9], thus offering new prospects for a possible use against other parasites, such as Diptera larvae.

The purpose of this study was to assess the *in vitro* larvicidal activity of ivermectin and PVP-I, alone and in combination, against $L_1$ *O. ovis* larvae.

## Materials and methods

### Pharmacological preparations

Two different formulations of ivermectin were tested. The first formulation was 1% ivermectin suspension in balanced sterile saline solution (BSSS). One-hundred mg of ivermectin powder (Sigma-Aldrich; ID PubChem: 24278497) was suspended in 10 mL of BSSS and placed on a magnetic stirrer for 5 minutes. The pH of the suspension was 5.3. The second formulation was 1% ivermectin solution in propylene glycol (propane-1,2-diol, $C_3H_8O_2$) vehicle. One-hundred mg of ivermectin powder (Sigma-Aldrich; ID PubChem: 24278497) was suspended in 5 ml of propylene glycol and placed on a magnetic stirrer for 5 minutes. After complete solubilization of ivermectin, the solution was brought to a volume of 10 mL by adding propylene glycol. The final pH was 5.4.

A combination of ivermectin 1% solution in propylene glycol and 0.6% PVP-I, obtained from 5% PVP-I ophthalmic solution (Oftasteril®, Alfa Intes, Casoria, Italy) was also tested. One-hundred mg of ivermectin powder was suspended in 5 mL of propylene glycol and placed on a magnetic stirrer for 5 minutes. Then, 1.2 mL of Oftasteril® was added and this solution was finally made up to 10 mL with propylene glycol. The final pH was 4.2.

Propylene glycol, the diluent used for ivermectin, was also tested alone, to avoid potential bias due to a possible larvicidal activity.

Finally, we assessed the larvicidal activity of a new commercial ophthalmic solution containing PVP-I 0.6% in hyaluronic acid vehicle (IODIM®, Medivis, Catania, Italy).

All the formulations were weighted on an analytical balance (Readability 0,1 mg, Mettler Toledo ab-204) and pH was measured with a pH-meter model 700 XS.

### *In vitro* larvicidal tests

*In vitro* experiments were performed at the Section of Parasitology and Parasitic Diseases, Department of Veterinary Medicine, University of Sassari, Sassari, Italy.

Larvae were collected from the nasal boots of sheep slaughtered in local certified abattoirs. Each day, a maximum of 15 heads from freshly slaughtered animals were obtained and transported in specific containers to the autopsy room of the Department of Veterinary Medicine. The sheep heads were cut into halves on the sagittal axis and the nasal boots were removed. Although every effort was made to process all the specimens on the slaughter day, when this was not feasible, the surplus heads were maintained intact in specific containers at 39°C and processed one day later.

All the nasal boots were examined by stereomicroscope. All the viable $L_1$ larvae found were collected and transferred onto separate Petri dishes containing mucosal tissue obtained from the nasal boots.

Each Petri dish was seeded with 25 larvae, filled with 1 mL of each study drug, and examined by light microscopy to assess larval viability. In order to quantify the time needed for each drug to kill the larvae, we defined *time of death* as the time necessary to obtain total larval immobility. When *time of death* was reached, the larvae were returned to drug-free culture medium and re-examined for a minimum of 5 minutes, in order to double-check their viability. If the absence of viability was confirmed, the *time of death* was accepted.

Control Petri dishes received 1 mL of BSSS.

All assays were performed in six replicates.

### Statistical analysis

Data were tested for normality (Shapiro Wilk and Kolmogorov-Smirnov test) and homogeneity of variance (Levine test). Furthermore, we performed a logarithmic (base 10) transformation of data, to make them closer to normal distribution and reduce skewness.

Due to non-parametric distribution and heterogeneity of variance, Kruskal Wallis H test and Dunnett's test with Sidák adjustment for multiple comparisons were performed on transformed data.

Kaplan-Meier survival analysis using the log-rank test was performed to compare the different drugs assessed.

A p value $< 0.05$ was considered to be statistically significant. Statistical analysis was carried out using Stata software (Stata/MP 14.1 for Mac, StataCorp, College Station, TX).

## Results

Only three sheep heads were held overnight in specific containers at 39°C and processed early in the morning the following day.

In total, 893 fully viable $L_1$ *O. ovis* larvae were tested in 50 days. The median time needed to kill the larvae was 96, 46, 44, 11, 6, and 10 minutes for BSSS, Iodim®, ivermectin 1% suspension in BSSS, propylene glycol, ivermectin 1% solution in propylene glycol, and a combination of ivermectin 1% solution in propylene glycol with 0.6% PVP-I, respectively (Table 1).

**Table 1. Median time required by Balanced Sterile Saline Solution (BSSS), Iodim®, ivermectin 1% suspension in BSSS, propylene glycol, ivermectin 1% solution in propylene glycol, and a combination of ivermectin 1% solution in propylene glycol with 0.6% povidone-iodine (PVP-I) to kill L1 *O. ovis* larvae.**

| Drug | Time | Median | Interquartile Range | Minimum | Maximum |
|---|---|---|---|---|---|
| **BSSS** | minutes | 94.35 | 132.35 | 6 | 253.53 |
| | Log10 | -3.753876 | -0.5485945 | -2.556303 | -4.182786 |
| **Iodim®** | minutes | 46.32 | 27.2 | 2.23 | 139.3 |
| | Log10 | -3.445915 | -0.2815163 | -2.155336 | -3.922725 |
| **Ivermectin 1% Suspension** | minutes | 44.21 | 17.36 | 6.26 | 64.28 |
| | Log10 | -3.425034 | -0.1974986 | -2.586587 | -3.587487 |
| **Propylene glycol** | minutes | 11.2 | 6.23 | 4.1 | 28.43 |
| | Log10 | -2.832188 | -0.2144022 | -2.39794 | -3.236285 |
| **1% Ivermectin Solution** | minutes | 6.2 | 3.6 | 1.19 | 12.59 |
| | Log10 | -2.579202 | -0.2913892 | -1.897627 | -2.891537 |
| **1% Ivermectin Solution + PVP-I 0.6%** | minutes | 10.13 | 4.21 | 2.8 | 16.18 |
| | Log10 | -2.787461 | -0.1931484 | -2.10721 | -2.990339 |

Results of Kaplan-Meyer survival analysis using the log-rank test are reported in Fig 1.

The survival curves were significantly lower in the samples treated with ivermectin 1% solution in propylene glycol, ivermectin 1% solution in propylene glycol + 0.6% PVP-I, and propylene glycol than in the samples receiving other treatments or BSSS.

Both original data and log10 transformations did not show a normal distribution; however, statistical analysis was performed using the latter. Kruskal-Wallis H test showed a statistically significant difference in *time of death* between the different drug treatments ($\chi^2[2] = 668.8$, with 5 d.f.; $p = 0.0001$). Dunnet test with Sidák adjustment for multiple comparisons showed that all drugs tested were significantly more effective than BSSS ($p < 0.001$).

Propylene glycol, the solvent chosen for ivermectin, showed a significantly higher larvicidal activity than Iodim® and ivermectin 1% suspension in BSSS ($p < 0.001$). However, ivermectin 1% solution in propylene glycol and the combination ivermectin 1% solution in propylene

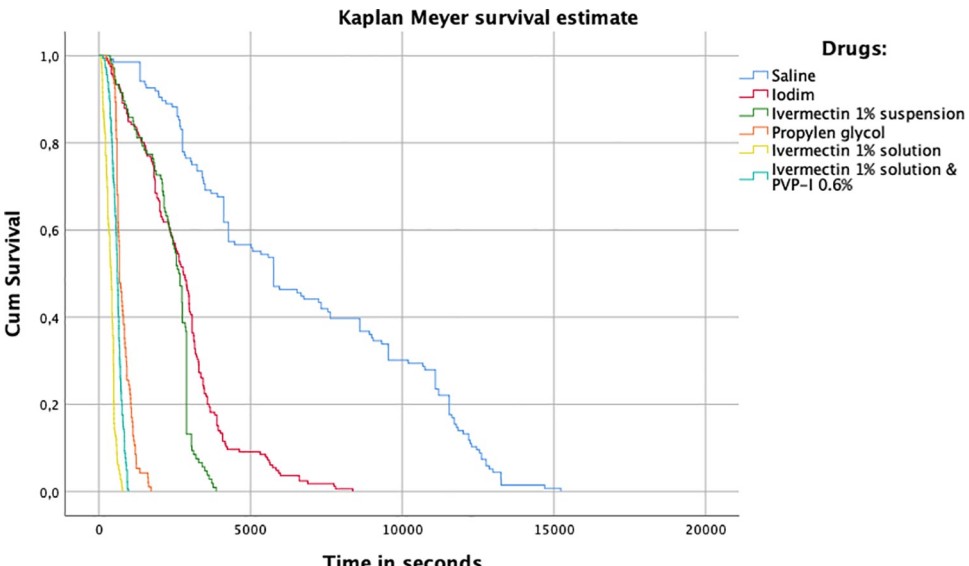

**Fig 1. Kaplan Meyer survival estimates.** The graph shows the survival curves of all the preparations tested.

glycol + PVP-I 0.6% were found to be significantly more effective than propylene glycol alone in terms of time needed to kill $L_1$ *O. ovis* larvae (p = 0.001 and p = 0.0422, respectively).

## Discussion

*O. ovis* larvae usually colonize the frontal and nasal sinuses of sheep and goat, but occasionally they can infest the human eyes [1–3, 10, 11] and naso-pharyngeal tract [12, 13]. *O. ovis* life cycle starts when a gravid fly sprays up to 25 larvae on the target animal face. Then, $L_1$ larvae migrate to the nasal boot using their hooks and feed on the mucus produced in response to the inflammatory process. Here, $L_1$ larvae stay for several months during winter (hypobiosis). When weather conditions become more favorable, normally during spring/summer, they start their metamorphosis through the $L_2$ and $L_3$ stages. $L_3$ larvae migrate out of the nasal boot and fall on the ground, where they develop into the adult form, thus starting a new life cycle.

Ivermectin, a macrocyclic lactone, is a semisynthetic anthelmintic agent for oral administration in the treatment of ascariasis, filariasis, gnathostomiasis, and hookworm infections [14]. Ivermectin binds selectively and with high affinity to glutamate-gated chloride ion channels in invertebrate muscle and nerve cells. This binding causes an increased cell membrane permeability to chloride ions and results in cell hyperpolarization, which leads to paralysis and death of the parasite. Furthermore, ivermectin also acts as a gamma-aminobutyric acid (GABA) agonist, thereby disrupting GABA-mediated neurosynaptic transmission. In general, ivermectin does not cross the blood-brain barrier in most animal species, including humans, and does not interact with peripheral neurotransmitters, making it safe enough for human use [15].

Oral ivermectin has been reported to be effective in the treatment of head and neck myiasis in humans [4, 16]. Furthermore, topical ivermectin showed a beneficial effect in the treatment of rosacea and blepharitis, conditions in which the mite *Demodex* is believed to play a role [17].

Povidone–iodine (PVI) is a disinfectant and antiseptic agent used for preoperative preparation of the skin and mucous membranes, as well as for the treatment of contaminated wounds. Because of its broad-spectrum antimicrobial activity, 5% PVI has been widely used in ophthalmology. A peculiar chemical property of povidone is that the concentration of free iodine, the active antimicrobial element, increases with the dilution of PVI, due to a weakening of the chemical bonding between iodine and povidone. Recently, a new ophthalmic preparation containing 0.6% PVI has been shown to be bactericidal against several Gram-positive and Gram-negative isolates [18]. Furthermore, 0.6% PVI has also been found to be effective against *Demodex mites* [8], a parasite colonizing the eyelid margins, thus offering new prospects for a possible use against other parasites, such as Diptera larvae.

In this *in vitro* study, we found that ivermectin 1% solution in propylene glycol vehicle and a combination of ivermectin 1% solution in propylene glycol with 0.6% PVP-I showed a good, rapid larvicidal activity against $L_1$ *O. ovis* larvae. Surprisingly enough, propylene glycol, the solubilizer chosen for ivermectin, had a similar larvicidal activity.

Ivermectin is insoluble and unstable in water but soluble in propylene glycol [19], a commonly used drug solubilizer in topical, oral, and injectable medications (e.g., intravenous diazepam, lorazepam, phenobarbital, phenytoin, and nitroglycerin). In Ophthalmology, propylene glycol is used in eye lubricants.

To the best of our knowledge, we are unaware of any former *in vitro* study assessing the larvicidal activity of ivermectin and PVP-I, alone and in combination, against $L_1$ *O. ovis* larvae. Our results suggest that ivermectin 1% solution in propylene glycol, ivermectin 1% solution in propylene glycol + 0.6% PVP-I, and propylene glycol might be potential candidates for the topical treatment, as ointment and/or eye-drops, for external ophthalmomyiasis caused by *O. ovis*.

Some authors have reported sporadic cases of human ophthalmomyiasis treated with a combination of petroleum ointment and ivermectin or ivermectin solution alone as adjuvant to the mechanical removal of the larvae. However, all these reports described cases of ophthalmomyiasis caused by other Diptera species (*Cochliomyia hominivorax* and *Dermatobia hominis*) [4–7].

There is already at least one paper indicating efficacy of oral ivermectin (12 mg in a single dose) in treating human conjunctival myiasis caused by *O. ovis* larvae [8]. However, the use of systemic ivermectin for the management of ocular surface myiasis is rather questionable, as a topical approach with ointment and/or eye-drops would be much more appropriate.

A clear limitation of our study is that we performed an *in vitro* experiment, which may not reflect exactly the real situation *in vivo*. Furthermore, we do not know how toxic ivermectin and propylene glycol can be for the corneal and conjunctival epithelium, at the concentrations used in this experiment. This topic will be the subject of future research. On the other hand, we feel strongly that our *in vitro* evaluation of different anthelmintic formulations on *O. ovis* viability represents a potentially impactful approach to addressing ophthalmomyiasis in the Mediterranean Basin or other sheep-rearing regions globally.

## Conclusions

We found that ivermectin 1% solution in propylene glycol vehicle, 1% ivermectin solution in propylene glycol + 0.6% PVP-I, and propylene glycol alone showed a good, relatively rapid larvicidal activity against $L_1$ *O. ovis* larvae. Further experimental and clinical studies are necessary to establish whether, or not, these formulations may be considered as potential candidates for the topical treatment for external ophthalmomyiasis caused by *O. ovis*.

## Acknowledgments

The authors wish to thank Dr Rosaria De Santis, an independent biologist, who provided writing support and critical analysis of the data.

## Author Contributions

**Conceptualization:** Giuseppe D'Amico Ricci, Giovanni Garippa, Stefano Cortese, Francesco Boscia, Antonio Pinna.

**Data curation:** Giuseppe D'Amico Ricci, Giovanni Garippa, Stefano Cortese, Antonio Pinna.

**Formal analysis:** Giuseppe D'Amico Ricci, Stefano Cortese, Rita Serra, Stefano Dore, Antonio Pinna.

**Funding acquisition:** Giovanni Garippa.

**Investigation:** Giuseppe D'Amico Ricci, Giovanni Garippa, Stefano Cortese.

**Methodology:** Giuseppe D'Amico Ricci, Giovanni Garippa, Stefano Cortese, Francesco Boscia.

**Supervision:** Giuseppe D'Amico Ricci, Giovanni Garippa, Stefano Cortese, Rita Serra, Stefano Dore, Antonio Pinna.

**Validation:** Giuseppe D'Amico Ricci, Giovanni Garippa, Stefano Cortese, Rita Serra, Francesco Boscia, Stefano Dore, Antonio Pinna.

**Writing – original draft:** Antonio Pinna.

**Writing – review & editing:** Giuseppe D'Amico Ricci, Giovanni Garippa, Stefano Cortese, Rita Serra, Francesco Boscia, Stefano Dore, Antonio Pinna.

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
