## [Decision Letter · Decision Letter 0]

6 Sep 2021

PONE-D-21-22404In vitro larvicidal activity of ivermectin and povidone-iodine against Oestrus ovisPLOS ONE

Dear Dr. Antonino Pinna,

Thank you for submitting your manuscript to PLOS ONE. After careful consideration, we feel that it has merit but does not fully meet PLOS ONE’s publication criteria as it currently stands. Therefore, we invite you to submit a revised version of the manuscript that addresses the points raised during the review process.

We look forward to receiving your revised manuscript.

Kind regards,

Filippo Giarratana

Academic Editor

PLOS ONE

Journal Requirements:

2. In your Methods section, please provide additional details regarding the ivermectin used in your study and ensure you have described the source. For more information regarding PLOS' policy on materials sharing and reporting, see https://journals.plos.org/plosone/s/materials-and-software-sharing#loc-sharing-materials.

Additional Editor Comments:

Dear author,

read carefully the comments of the Reviewer 2 and response to all the suggestion/requirement.

Reviewers' comments:

Reviewer's Responses to Questions

**Comments to the Author**

1. Is the manuscript technically sound, and do the data support the conclusions?

Reviewer #1: Yes

Reviewer #2: Yes

2. Has the statistical analysis been performed appropriately and rigorously? 

Reviewer #1: Yes

Reviewer #2: I Don't Know

3. Have the authors made all data underlying the findings in their manuscript fully available?

Reviewer #1: Yes

Reviewer #2: Yes

4. Is the manuscript presented in an intelligible fashion and written in standard English?

Reviewer #1: Yes

Reviewer #2: Yes

5. Review Comments to the Author

Reviewer #1: The results described in the manuscript are preliminary, however they are very interesting and provide the basis for clinical studies. The only negative point, but not an impediment to publication, is that the authors should have performed the in vitro assays with a greater number of repetitions.

Reviewer #2: Re: PLOS One Manuscript PONE-D-21-22404

In vitro larvicidal activity of ivermectin and povidone-iodine against Oestrus ovis

By Giuseppe D’Amico Ricci

General:

The authors have submitted a well prepared manuscript describing laboratory effectiveness of five formulations, containing one or more of three main components, the parasiticide ivermectin, propylene glycol, used mainly as a carrier, but also tested as a sole ingredient as a control, and povidone-iodine in hyaluronic acid vehicle which is used primarily as an antiseptic but with a previous report of insecticidal activity against Demodex mites noted by the authors.

Usually laboratory testing, such as conducted here, is conducted as a preliminary study to in vivo trials to establish whether any insecticidal action against the target organism is present. If so in vivo effectiveness is validated later. Ivermectin often acts systemically and lack of a topical effect in laboratory studies does not necessarily mean that it would not be effective in vivo and provides very little information about optimal formulation.

However, in this instance there is already at least one paper indicating efficacy of ivermectin in treating human ocular myiasis caused by Oestrus ovis larvae (Kumar et al. ), some studies indicating efficacy of ivermectin against O oestrus in human nasal infestations and a significant number of papers indicating in vivo activity against O. ovis in sheep which draws into question the novelty of the work and need for new laboratory studies to demonstrating insecticidal effect against O. ovis. Furthermore, it could perhaps be argued from the results that the polypropylene glycol was more active in the laboratory studies than ivermectin per se, although whether this would be the case in a clinical setting is questionable.

Methods:

More detail is needed to for the reader to accurately interpret the design used for the study. It is noted that “Each Petri dish was seeded with 25 larvae, filled with 1 mL of each study drug” and that all assays were conducted in triplicate. I may have misunderstood, but to me this suggests that 5 treatments + saline control x 25 larvae x 3 replicates = 450 larvae were used. However, it is mentioned in the abstract and in the body of the paper that 893 larvae were tested. How were the other larvae used? This needs to be clarified

It is noted that larvae for the study were collected over a number of days and that in most cases the experiments were conducted on the day of collection, but in some instances when all heads could not be processed, they were held overnight and processed the following day. Often such closely associated parasites die quickly once separated from their host, or their host dies. Thus how the replicates/treatments were established with respect to time of collection needs to be specified to ensure there is no confounding between treatment effects and time of collection.

Results.

Figure 1.

As I understand it Figure 1, Graph A contains a plot of all treatments, Graph B is exactly the same as A but with 3 of the treatments from Graph A omitted and Graphs C and D just contain subsets of Graph A replotted with a different scale on the X axis?. That is, for example the same data 1% ivermectin in BSSS treatment is replotted 3 times in Graphs A, B, and C. It is unusual to replot the same data repeatedly and is potentially confusing for the reader. Consider just using Graph A with statements about which treatments were significantly different in the caption.

References.

As noted above there is at least one previous reported study of the use of ivermectin to treat human ocular myiasis (Kumar et al. 2013 Annals of Tropical Medicine and Public Health 6: 315-316) as well as a quite a number of previous reports of the effectiveness of ivermectin against Oestrus ovis larvae (in human hosts and in sheep). None of these papers have been cited in the reference list. These papers would seem to be very relevant to the current study and at least some of them should be noted.

6. PLOS authors have the option to publish the peer review history of their article (what does this mean?). If published, this will include your full peer review and any attached files.

Reviewer #1: No

Reviewer #2: No

---

## [Author Response · Author response to Decision Letter 0]

14 Sep 2021

Journal requirements

- Please ensure that your manuscript meets PLOS ONE's style requirements, including those for file naming

Reply: done

- In your Methods section, please provide additional details regarding the ivermectin used in your study and ensure you have described the source.

Reply: additional details regarding the ivermectin used in our study have been provided.

Reviewer #1

- The results described in the manuscript are preliminary, however they are very interesting and provide the basis for clinical studies. The only negative point, but not an impediment to publication, is that the authors should have performed the in vitro assays with a greater number of repetitions.

Reply: All assays were performed in six replicates, not in triplicate. Triplicate was a misprint

Reviewer #2

- More detail is needed to for the reader to accurately interpret the design used for the study. It is noted that “Each Petri dish was seeded with 25 larvae, filled with 1 mL of each study drug” and that all assays were conducted in triplicate. I may have misunderstood, but to me this suggests that 5 treatments + saline control x 25 larvae x 3 replicates = 450 larvae were used

Reply: We wish to thank Reviewer #2 for highlighting our mistake. Indeed, there was a misprint. All assays were performed in six replicates, not in triplicate (5 treatments + saline control x 25 larvae x 6 replicates = ~900 larvae were used)

- It is noted that larvae for the study were collected over a number of days and that in most cases the experiments were conducted on the day of collection, but in some instances when all heads could not be processed, they were held overnight and processed the following day. Often such closely associated parasites die quickly once separated from their host, or their host dies. Thus how the replicates/treatments were established with respect to time of collection needs to be specified to ensure there is no confounding between treatment effects and time of collection.

Reply: Only three sheep heads were held overnight in specific containers at 39°C and processed early in the morning the following day. This procedure did not significantly affect larval viability. All the nasal boots were examined by stereomicroscope. All the viable L1 larvae found were collected and transferred onto separate Petri dishes containing mucosal tissue obtained from the nasal boots.

- Figure 1.

As I understand it Figure 1, Graph A contains a plot of all treatments, Graph B is exactly the same as A but with 3 of the treatments from Graph A omitted and Graphs C and D just contain subsets of Graph A replotted with a different scale on the X axis?. That is, for example the same data 1% ivermectin in BSSS treatment is replotted 3 times in Graphs A, B, and C. It is unusual to replot the same data repeatedly and is potentially confusing for the reader. Consider just using Graph A with statements about which treatments were significantly different in the caption.

Reply: Figure 1 has been changed according to Reviewer #2 suggestions.

-References.

As noted above there is at least one previous reported study of the use of ivermectin to treat human ocular myiasis (Kumar et al. 2013 Annals of Tropical Medicine and Public Health 6: 315-316) as well as a quite a number of previous reports of the effectiveness of ivermectin against Oestrus ovis larvae (in human hosts and in sheep). None of these papers have been cited in the reference list. These papers would seem to be very relevant to the current study and at least some of them should be noted.

Reply: the study by Kumar et al. has been quoted (Reference 8) and additional references have been added (References 11, 12 and 13).

The following comment has been included in the Discussion: “There is already at least one paper indicating efficacy of oral ivermectin (12 mg in a single dose) in treating human conjunctival myiasis caused by O. ovis larvae.8 However, the use of systemic ivermectin for the management of ocular surface myiasis is rather questionable, as a topical approach with ointment and/or eye-drops would be much more appropriate.”

---

## [Decision Letter · Decision Letter 1]

6 Oct 2021

PONE-D-21-22404R1In vitro larvicidal activity of ivermectin and povidone-iodine against Oestrus ovisPLOS ONE

Dear Dr. Antonio Pinna,

Thank you for submitting your manuscript to PLOS ONE. After careful consideration, we feel that it has merit but does not fully meet PLOS ONE’s publication criteria as it currently stands. Therefore, we invite you to submit a revised version of the manuscript that addresses the points raised during the review process.

ACADEMIC EDITOR:

Please improve the paper with the suggestion of the Reviewer #2.

We look forward to receiving your revised manuscript.

Kind regards,

Filippo Giarratana

Academic Editor

PLOS ONE

Journal Requirements:

Reviewers' comments:

Reviewer's Responses to Questions

**Comments to the Author**

1. If the authors have adequately addressed your comments raised in a previous round of review and you feel that this manuscript is now acceptable for publication, you may indicate that here to bypass the “Comments to the Author” section, enter your conflict of interest statement in the “Confidential to Editor” section, and submit your "Accept" recommendation.

Reviewer #1: All comments have been addressed

Reviewer #2: All comments have been addressed

2. Is the manuscript technically sound, and do the data support the conclusions?

Reviewer #1: Yes

Reviewer #2: Yes

3. Has the statistical analysis been performed appropriately and rigorously? 

Reviewer #1: Yes

Reviewer #2: Yes

4. Have the authors made all data underlying the findings in their manuscript fully available?

Reviewer #1: Yes

Reviewer #2: Yes

5. Is the manuscript presented in an intelligible fashion and written in standard English?

Reviewer #1: Yes

Reviewer #2: Yes

6. Review Comments to the Author

Reviewer #1: Authors respond to reviewers' comments in a satisfactory manner, in this way the manuscript can be accepted as is.

Reviewer #2: Re: PLOS One Manuscript PONE-D-21-22404-R1

In vitro larvicidal activity of ivermectin and povidone-iodine against Oestrus ovis

The manuscript has been revised appropriately and with some minor suggested amendments noted below should be suitable for publication in PLOS 1.

Abstract

Lines 39-41 “Kaplan-Meyer analysis

…disclosed that the survival curves were significantly lower in the samples treated with ivermectin 41 1% solution, ivermectin 1% solution + 0.6% PVP-I, and propylene glycol.” Please state lower than what. Lower than the RSSS control or lower than the other treatments

Introduction

Line 54. In entomological protocol it is most common, though not essential to state both the Order and the Family e.g. [Diptera:Oestridae]

Line 115- The BSS would usually be considered a negative control. A positive control would usually mean a treatment known to be effective in killing larvae. I suggest just delete “Positive”: ie Control Petri dishes received 1 mL of BSSS.

Line 124 – Log rank test, rather than “..long rank test”

Line 147 – Again, as above, significantly lower than what?

Lines 175-177. This is an important point which would benefit from a supporting reference.

Line 193 to 195 Delete “also”. That is: ‘Surprisingly enough, propylene glycol, the solubilizer chosen for ivermectin, had similar larvicidal activity.

Lines 203-204 Suggest “Our results suggest that ivermectin 1% solution in propylene glycol, ivermectin 1% solution in propylene glycol + 0.6% PVP-I, and propylene glycol might be potential candidates for the topical treatment, as ointment and/or eye-drops, for external ophthalmomyiasis caused by O. ovis.”

Line 224 – closing sentence. Suggest ‘… to establish whether or not these formulations may be considered….’ Rather than “…they..”

7. PLOS authors have the option to publish the peer review history of their article (what does this mean?). If published, this will include your full peer review and any attached files.

Reviewer #1: No

Reviewer #2: No

---

## [Author Response · Author response to Decision Letter 1]

11 Oct 2021

Journal requirements

Reply: As far as we know, there are no retracted references in the list. Reference 8 (Kumar MA, Joseph NM, Srikanth K, Stephen S. Conjunctival Myiais caused by Oestrus ovis in a medical college student which responded to Ivermectin. Ann Trop Med Public Health 2013;6: 315-316), whose inclusion in the list was recommended by Reviewer #2, does not appear in PubMed, but can be retrieved in Scopus 

Reviewer #2

Abstract

Lines 39-41 “Kaplan-Meyer analysis

…disclosed that the survival curves were significantly lower in the samples treated with ivermectin 41 1% solution, ivermectin 1% solution + 0.6% PVP-I, and propylene glycol.” Please state lower than what. Lower than the RSSS control or lower than the other treatments

Reply: the sentence has been changed as follows: Kaplan-Meyer analysis disclosed that the survival curves were significantly lower in samples treated with ivermectin 1% solution, ivermectin 1% solution + 0.6% PVP-I, and propylene glycol than in samples receiving other treatments or BSSS.

Introduction

Line 54. In entomological protocol it is most common, though not essential to state both the Order and the Family e.g. [Diptera:Oestridae]

Reply: changed according to Reviewer suggestion

Line 115- The BSS would usually be considered a negative control. A positive control would usually mean a treatment known to be effective in killing larvae. I suggest just delete “Positive”: ie Control Petri dishes received 1 mL of BSSS.

Reply: changed according to Reviewer suggestion

Line 124 – Log rank test, rather than “..long rank test”

Reply: changed according to Reviewer suggestion

Line 147 – Again, as above, significantly lower than what?

Reply: the sentence has been changed as follows: The survival curves were significantly lower in the samples treated with ivermectin 1% solution in propylene glycol, ivermectin 1% solution in propylene glycol + 0.6% PVP-I, and propylene glycol than in the samples receiving other treatments or BSSS.

Lines 175-177. This is an important point which would benefit from a supporting reference.

Reply: as recommended the following reference (#15) has been included in the list: Laing R, Gillan V, Devaney E. Ivermectin - Old Drug, New Tricks? Trends Parasitol. 2017;33: 463-472.

Line 193 to 195 Delete “also”. That is: ‘Surprisingly enough, propylene glycol, the solubilizer chosen for ivermectin, had similar larvicidal activity.

Reply: changed according to Reviewer suggestion

Lines 203-204 Suggest “Our results suggest that ivermectin 1% solution in propylene glycol, ivermectin 1% solution in propylene glycol + 0.6% PVP-I, and propylene glycol might be potential candidates for the topical treatment, as ointment and/or eye-drops, for external ophthalmomyiasis caused by O. ovis.”

Reply: changed according to Reviewer suggestion

Line 224 – closing sentence. Suggest ‘… to establish whether or not these formulations may be considered….’ Rather than “…they..”

Reply: changed according to Reviewer suggestion

---

## [Editor Report · Decision Letter 2]

12 Oct 2021

In vitro larvicidal activity of ivermectin and povidone-iodine against Oestrus ovis

PONE-D-21-22404R2

Dear Dr. Antonio Pinna,

We’re pleased to inform you that your manuscript has been judged scientifically suitable for publication and will be formally accepted for publication once it meets all outstanding technical requirements.

Kind regards,

Filippo Giarratana

Academic Editor

PLOS ONE

---

## [Editor Report · Acceptance letter]

14 Oct 2021

PONE-D-21-22404R2 

In *vitro* larvicidal activity of ivermectin and povidone-iodine against *Oestrus ovis*

Dear Dr. Pinna:

I'm pleased to inform you that your manuscript has been deemed suitable for publication in PLOS ONE. Congratulations! Your manuscript is now with our production department. 

Kind regards, 

on behalf of

Dr. Filippo Giarratana 

Academic Editor

PLOS ONE